# Feasibility of Early Evaluation for the Recurrence of Bladder Cancer after Trans-Urethral Resection: A Comparison between Magnetic Resonance Imaging and Multidetector Computed Tomography

**Yiqian Wang** [1,2,†] **, Wei Zhang** [1,2,†] **, Weixiong Xiao** [1,2] **, Shaobin Chen** [1,2] **, Yongbao Wei** [1,3] **and Min Luo** [1,2,*]

[1] Shengli Clinical Medical College of Fujian Medical University, Fuzhou 350001, China
[2] Department of Radiology, Fujian Provincial Hospital, Fuzhou 350001, China
[3] Department of Urology, Fujian Provincial Hospital, Fuzhou 350001, China
* Correspondence: luomin6668@163.com
† These authors contributed equally to this work.

**Abstract:** (1) Background: This study investigates the early evaluation value of magnetic resonance imaging (MRI) and multidetector computed tomography (MDCT) in diagnosing the recurrence of bladder cancer (BC) after trans-urethral resection (TUR) alone or combined with intravesical perfusion chemotherapy. (2) Methods: This retrospective study enrolled 92 patients with BC who underwent MRI and MDCT after TUR. The time interval between MRI and MDCT was no more than 1 week. Tumor recurrence was recorded by two experienced radiologists who were double-blind. Recurrent patients were divided into nodular masses, irregular wall thickening and smooth wall thickening groups according to tumor morphology in cystoscopy and resected gross specimens. Inter- and intra-observer agreement was evaluated using the Kappa test. Imaging diagnostic performance was assessed using receiver operating characteristic (ROC) analysis and McNemar's test based on pathology. (3) Results: There were 56 relapsed and 36 non-relapsed patients. The intra-observer agreement for the imaging diagnosis was excellent ($\kappa$ = 0.96 for MRI and $\kappa$ = 0.91 for MDCT, both $p < 0.001$). The area under the ROC curve of MRI was higher than that for MDCT (0.91 vs. 0.74, $p < 0.001$) in identifying tumor recurrence and benign treatment-related changes. The sensitivity, specificity and accuracy of MRI (87.5%, 94.4% and 90.2%, respectively) were higher than those of MDCT (67.9%, 80.6% and 72.8%, respectively) in diagnosing tumor recurrence. Two observers missed 10 cases of small lesions (<1 cm) on MDCT. The accuracy of MRI (100%, 90.0% and 25.0%, respectively) was higher than that of MDCT (92.1%, 30.0% and 0%, respectively) in diagnosing nodular masses, irregular wall thickening and smooth wall thickening recurrence patterns. (4) Conclusions: Compared with MDCT, MRI had a higher accuracy in detecting BC recurrence early, especially for nodular masses and irregular wall thickening, and could better differentiate tumor recurrence from benign treatment-related changes.

**Keywords:** bladder cancer; trans-urethral resection; recurrence; magnetic resonance imaging; multidetector computed tomography

## 1. Introduction

Bladder cancer (BC) is the most common malignancy of the urinary tract and has a high local recurrence rate. A combined analysis of 2596 patients with stage Ta-T1 BC who underwent trans-urethral resection of bladder tumor (TURBT) showed a one-year recurrence rate of 15–61% and a five-year recurrence rate of 31–78% [1]. For patients with muscle-invasive BC, radical cystectomy with orthotopic ileal neobladder or TURBT combined with intravesical perfusion therapy can eliminate most lesions, but 5–50% of patients experience local or metastatic recurrence within 24 months after the operation [2]. Even

after complete response, the bladder remains a potential source of recurrence. Therefore, postoperative follow-up is required to detect tumor recurrence early. The "early" refers to the first postoperative recurrence of BC in our study.

Follow-up is recommended every 3–6 months for the first 2 years after TURBT, every 6 months up to 5 years, and annually thereafter, and mainly includes cystoscopy and urine cytology [3]. Radiological assessment of the urinary tract is performed annually or in cases of tumor recurrence or suspicion. Cystoscopy is the gold standard for postoperative follow-up. However, owing to its invasiveness accompanied by the associated risks of hematuria or infection [4], only allowing the surface of the bladder mucosa to be observed, which is costly and time-consuming, limits the need for frequent postoperative use. Urine cytology is specific but its overall sensitivity is low and cannot accurately predict muscle-invasive or high-grade tumors [5,6]. Therefore, finding non-invasive techniques to accurately monitor the postoperative curative effect of BC is essential for the prognosis and management of patients.

Magnetic resonance imaging (MRI), computed tomography (CT) and ultrasound are convenient non-invasive imaging tools that can be repeated during follow-up, and allow visualization of the entire bladder and surrounding structures. In the past decades, bladder tumors have mainly been examined by CT or ultrasound. CT is the most common and convenient imaging modality because of its fast scanning speed, low cost, wide application, and accurate assessment of lymph nodes or distant metastases. CT urography can also detect the presence of upper urinary tract diseases. However, it is difficult to assess tumor muscle invasion and detect flat or small lesions because of its relatively poor tissue contrast [7,8]. CT also has ionizing radiation. Moreover, scar tissue generates after TURBT and inflammatory wall thickening occurs following intravesical perfusion therapy, which are often difficult to differentiate from tumor recurrence by CT and ultrasound.

In recent years, MRI has shown promising applications in the bladder. It has the advantages of superior soft tissue contrast, multiparametric imaging, and no ionizing radiation, which can better evaluate the soft tissue and pelvic structures, and may be more sensitive to bladder wall structures, local recurrence and bone diseases [9]. Multiparametric MRI (mpMRI) combining different scan sequences can improve diagnostic performance. Previous studies [10,11] indicated that: (1) mpMRI may be a complementary and potential alternative to cystoscopy, and could be used for follow-up and recurrence monitoring of BC; and (2) diffusion-weighted imaging (DWI), apparent diffusion coefficient (ADC) and dynamic contrast-enhanced (DCE) sequences could differentiate postoperative chronic inflammatory fibrosis from tumor recurrence.

To date, there have been no comparative reports of the use of MRI and CT to monitor the curative effect of TURBT. This is the first study to investigate the feasibility of differentiating tumor recurrence from benign treatment-related changes between MRI and multidetector CT (MDCT) in patients with BC after TURBT. This study aimed to compare the early evaluation value of MRI and MDCT in diagnosing the recurrence of BC after TURBT alone or combined with intravesical perfusion chemotherapy.

## 2. Materials and Methods

This retrospective study was conducted in accordance with the Declaration of Helsinki and written informed consent was obtained from all patients. This study was approved by the local Institutional Review Board.

### 2.1. Subjects

BC patients who underwent postoperative MRI and MDCT at our hospital between January 2014 and March 2022 were collected. Some patients had hematuria symptoms, and some patients were routinely followed up. Pelvic MRI is mainly used to check the bladder, and whole abdominal MDCT is mainly used to evaluate upper urinary tract abnormalities and metastases to retroperitoneum, abdominal organs or bone. Inclusion criteria were (1) a definite medical history of BC; (2) MRI and MDCT were performed at least 3 months

after TURBT; (3) the time interval between MRI and MDCT was no more than 1 week, and cystoscopy or other clinical operations were not performed during this period; and (4) underwent surgery or cystoscopy biopsy after imaging examination. Exclusion criteria were (1) patients with previous history of BC recurrence (n = 36); and (2) poor image quality (n = 2). Finally, the cohort comprised 92 patients. It was found that all patients included in the study were free of lymph node or distant metastasis before this imaging examination.

### 2.2. Patient Preparation

One hour before examination, after the urine was emptied, patients drank 300–500 mL water to make the bladder well filled. Bladder filling was examined on the localizer image and examinations were delayed if the bladder was not filled. In addition, the patients were instructed to defecate 6 h before the MRI examination to reduce intestinal gas interference.

### 2.3. Equipment and Parameters

MRI and MDCT were performed using the 3.0-T MR scanner (Siemens MAGNETOM Prisma, Erlangen, Germany) and 64-row CT scanner (Siemens Sensation, Erlangen, Germany).

MRI received signals through a 32-channel spine coil and an 18-channel phased-array body coil with the patients in the supine position. The scan ranges from the upper margin of the ilium to the symphysis pubis. MRI protocols included T1-weighted imaging, fast spin-echo T2-weighted imaging, free-breathing DWI with single-shot echo-planar imaging and fat-suppressed 3D T1-weighted volumetric interpolated breath-hold examination (T1-VIBE) DCE images (Table 1). The contrast agent was injected with 0.2 mmol/kg of Gd-DTPA (Bayer HealthCare Pharmaceuticals, Leverkusen, Germany) at a rate of 2 mL/s, through the dorsal hand vein with a power-injector system, followed by a 15–20 mL saline flush at the same rate. Unenhanced scanning was performed before injection, and 11 sets of axial contrast-enhanced images were acquired, followed by delayed scanning in the sagittal, coronal and axial planes.

**Table 1.** 3.0-T MRI protocol parameters.

| Parameter | T1WI | T2WI | DWI | DCE |
|---|---|---|---|---|
| plane | axial | axial, sagittal and coronal | axial and sagittal (or coronal) | axial, sagittal and coronal |
| echo time (ms) | 2.78/1.36 (in/opp) | 85 | 50 | 1.31 |
| repetition time (ms) | 4.48 | 4280 | 7000 | 3.16 |
| field of view (mm$^2$) | 380 × 308 | 240 × 240 | 240 × 211 | 280 × 280 |
| matrix | 182 × 320 | 275 × 320 | 100 × 88 | 224 × 320 |
| slice thickness (mm) | 3.5 | 3.5 | 3.5 | 3.5 |
| slice gap (mm) | 0.4 | 0.4 | 0.4 | 0.4 |
| b-value (s/mm$^2$) | | | 50, 600, 1200 | |
| average | 1 | 1 | 2, 3, 4 | 1 |
| total acquisition time | 16 s | 5 min 2 s | 7 min 28 s | 5 min 45 s |

MRI, magnetic resonance imaging; T1WI, T1-weighted imaging; T2WI, T2-weighted imaging; DWI, diffusion-weighted imaging; DCE, dynamic contrast-enhanced.

The MDCT parameters were as follows: tube voltage, 120 kV; tube current, 230 mA; rotating speed 0.5 s/r; pitch, 1.1; collimation, 64 × 0.6 mm; convolution kernel, B30f; reconstruction slice thickness and interval were all 3 mm. A total of 1.5 mL/kg of non-ionic contrast media (ioversol, 320 mg·I/mL, Jiangsu Hengrui Pharmaceutical Co., Ltd, Jiangsu, China, or iopamidol, 300 mg·I/mL, Beijing Beilu Pharmaceutical Co., Ltd, Beijing, China) and 15 mL saline were injected intravenously at a rate of 3 mL/s using a computer-controlled injector. The corticomedullary, nephrographic, excretory and bladder filling phases were scanned at 25 s, 70 s, 3–5 min and 30 min after the injection, respectively. Mul-

tiplanar images were reconstructed on post-processing workstation (Syngo.via, Siemens Healthcare, Erlangen, Germany).

### 2.4. Image Analysis

The diagnostic criteria of MDCT was that nodular lesions or asymmetric wall thickening with greater enhancement than the adjacent bladder wall was considered tumor recurrence. Wall thickening without abnormal enhancement was considered to be inflammation.

For MRI, lesions show high or slightly high signal intensity (SI) on DWI, low or slightly low ADC values, early enhancement, and wash-out in delayed phase were regarded as recurrent tumors. Lesions that were low to equal SI on DWI, intermediate to high ADC values, and progressive enhancement were regarded as benign inflammation or fibrosis. According to the study of Li [12], we set $(0.72 \sim 0.90) \times 10^{-3}$ mm$^2$/s as the critical range of intermediate ADC values.

Lymph node was considered metastasis if the short diameter $\geq 10$ mm or ring enhancement. Metastasis of other organs was diagnosed with reference to their respective imaging characteristics.

All images were read by two radiologists (observers 1 and 2 with 7 and 9 years of experience in diagnosing pelvic diseases, respectively), who were double-blind (they knew the patients were postoperative with BC but were unaware of the histopathological findings). When diagnosis opinions of the same examination were different between observers, they reviewed examinations simultaneously and disagreements were resolved through consensus. MRI and MDCT were read at an interval of one month to reduce interference between images' evaluation. All images were reviewed again after three months by repeating the above process to assess intra-observer agreement. The observers recorded the lesion number and size simultaneously.

### 2.5. Reference Standard

All patients underwent surgery or cystoscopy biopsy within 1–4 weeks after the imaging examinations, and specimens were sent for pathological examination. Patients were divided into recurrence and non-recurrence groups according to the pathological results.

Recurrent patients were divided into nodular masses, irregular wall thickening and smooth wall thickening groups [13], stratified according to tumor morphology in cystoscopy and resected gross specimens. Nodular masses referred to lesions with papillary or cauliflower shape; irregular wall thickening referred to lesions that were focal wall thickening compared with the adjacent bladder wall, or diffuse wall thickening in cystoscopy, and the thicknesses of resected gross specimens were greater than or equal to 3 mm; and smooth wall thickening referred to lesions with smooth or slightly gross mucosa in cystoscopy, and the thicknesses of resected gross specimens were less than 3 mm.

### 2.6. Statistical Analysis

Statistical analysis was performed using the SPSS software (version 26.0) (IBM, Chicago, IL, USA). Quantitative data were described as mean $\pm$ standard deviation for normal distribution, median and interquartile range for non-normal distribution. Inter- and intra-observer agreement was evaluated using Kappa test, and $\kappa \leq 0.20$, 0.21–0.40, 0.41–0.60, 0.61–0.80 and >0.80 indicate slight, fair, moderate, good and excellent consistency, respectively. Receiver operating characteristic (ROC) analysis was used to evaluate the diagnostic performance of MRI and MDCT in distinguishing tumor recurrence from benign treatment-related changes. The difference between the imaging diagnosis and the reference standard was determined using McNemar's test. Sensitivity, specificity, false-positive rate, false-negative rate and accuracy were calculated. A $p$-value < 0.05 was considered statistically significant.

## 3. Results

### 3.1. Clinical Characteristics

The study included 92 patients (56 relapsed and 36 non-relapsed), 62 of whom received intravesical perfusion chemotherapy. The follow-up time was 3–99 months from the first operation to imaging examination. The flowchart of the study design is shown in Figure 1 and the clinical characteristics of the patients are summarized in Table 2.

In the recurrence group, all cases were pathologically confirmed as urothelial carcinoma (21 of G1, 4 of G2, 26 of G3, 3 of low-grade malignant potential urothelial papillary tumor, 1 of lymph node metastasis and 1 of urethral metastasis). In the non-recurrence group, 34 cases were inflammation and 2 cases were urothelial papilloma.

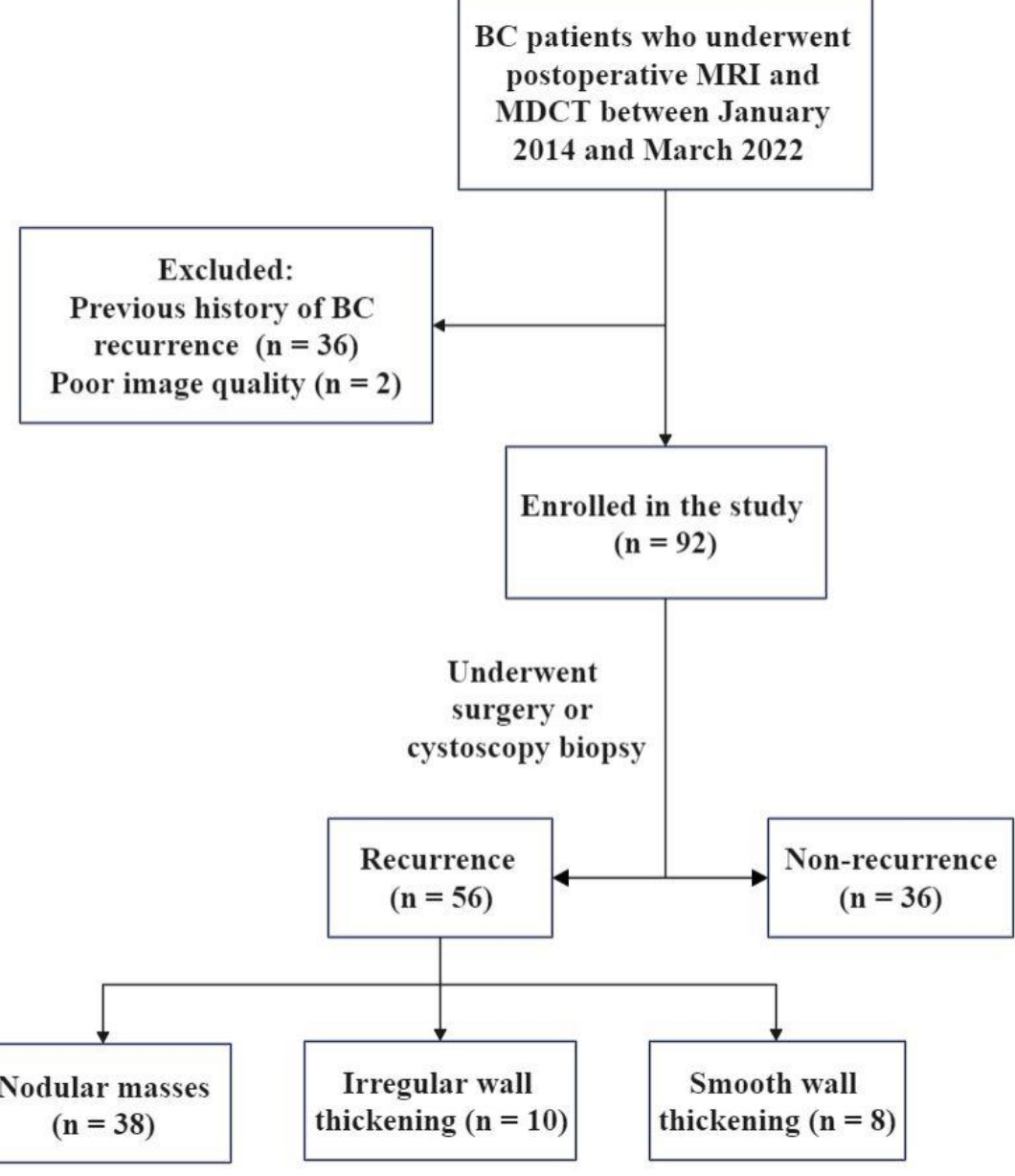

**Figure 1.** The flowchart of study design. BC, bladder cancer; MDCT, multidetector computed tomography.

**Table 2.** Clinical characteristics of postoperative bladder cancer patients.

| Variables | Total | Recurrence | Non-Recurrence |
|---|---|---|---|
| Case, n (%) | 92 | 56 (60.9) | 36 (39.1) |
| Age, mean ± SD (range), years | 66.5 ± 12.1 (29–89) | 68.0 ± 11.6 (43–89) | 64.2 ± 12.6 (29–83) |
| Gender, n (%) | | | |
| Male | 77 (83.7) | 47 (83.9) | 30 (83.3) |
| Female | 15 (16.3) | 9 (16.1) | 6 (16.7) |
| Follow-up time, median (IQR), months | 25.5 (11–50.5) | 35 (18.8–65.2) | 15 (5–27.5) |
| Tumor size, n (%) | | | |
| ≥1 cm | 49 (53.3) | 37 (66.1) | 12 (33.3) |
| <1 cm | 43 (46.7) | 19 (33.9) | 24 (66.7) |
| Lesion multiplicity | | | |
| Solitary | 58 (63.0) | 30 (53.6) | 28 (77.8) |
| Multiple | 34 (37.0) | 26 (46.4) | 8 (22.2) |
| Intravesical perfusion therapy | | | |
| Yes | 62 (67.4) | 35 (62.5) | 27 (75.0) |
| No | 30 (32.6) | 21 (37.5) | 9 (25.0) |

SD, standard deviation; IQR, interquartile range. Follow-up time, from the first operation to this imaging examination. Tumor size, indicates the largest tumor diameter measured on magnetic resonance imaging if multifocal tumor appeared.

### 3.2. Imaging Diagnostic Performance

The inter-observer agreement of the imaging diagnosis was excellent (κ = 0.91 for MRI and κ = 0.83 for MDCT, both $p < 0.001$). The intra-observer agreement was also excellent (κ = 0.96 for MRI and κ = 0.91 for MDCT, both $p < 0.001$).

The area under the ROC curve (Figure 2) for identifying tumor recurrence and benign treatment-related changes on MRI (0.91 [95% confidence interval (CI): 0.84–0.98]) was higher than that of MDCT (0.74 [95% CI: 0.64–0.85], $p < 0.001$).

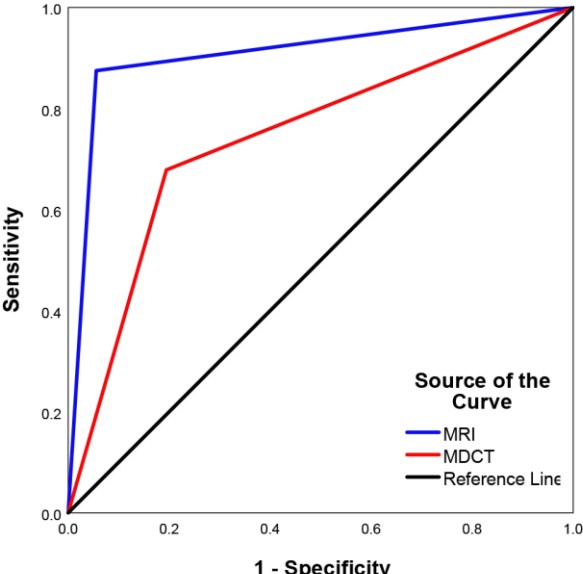

**Figure 2.** Receiver operating characteristic curve of MRI and MDCT for discriminating BC recurrence and benign treatment-related diseases.

The imaging diagnostic performance is shown in Table 3. There was no significant difference between the MRI diagnosis and pathological results ($p = 0.18$), while there was a significant difference between the MDCT diagnosis and pathological results ($p = 0.04$). A total of 167 tumors were found on MRI and 143 on MDCT. Based on histopathological findings, 10 cases of small lesions (<1 cm) were missed on MDCT by two observers (Figures 3 and 4).

One case of obvious diffuse wall thickening and one case of irregular wall thickening with patchy protrusion into the bladder were misdiagnosed as BCs by MRI and MDCT, and were pathologically confirmed as inflammation. One case of papilloma and the other four cases of irregular wall thickening were misdiagnosed as recurrences by MDCT.

The morphological classification of the recurrent lesions is summarized in Table 4. For nodular masses, MDCT missed one case of urethral metastasis and two cases of small lesions (0.2–0.5 cm). Among irregular wall thickening, six cases of focal wall thickening were diagnosed as inflammation on MDCT (Figure 5). These lesions showed low SI on T2-weighted imaging, high SI on DWI, intermediate or low ADC values, early obvious enhancement and slightly decreased enhancement in delayed phase on DCE-MRI. One case of diffuse wall thickening was diagnosed as glandular cystitis on MRI and MDCT. All these were confirmed pathologically as BCs. The accuracy of smooth wall thickening was the lowest, and all incorrect cases were diagnosed as postoperative inflammatory fibrous hyperplasia or negative findings.

**Table 3.** Diagnostic performance for recurrent BCs between MRI and MDCT.

|  | MRI | MDCT |
| --- | --- | --- |
| Sensitivity | 87.5% (49/56) | 67.9% (38/56) |
| Specificity | 94.4% (34/36) | 80.6% (29/36) |
| False-positive rate | 5.6% (2/36) | 19.4% (7/36) |
| False-negative rate | 12.5% (7/56) | 32.1% (18/56) |
| Accuracy | 90.2% (83/92) | 72.8% (67/92) |
| $x^2$ | 1.78 | 4.00 |
| $p$ | 0.18 | 0.04 |

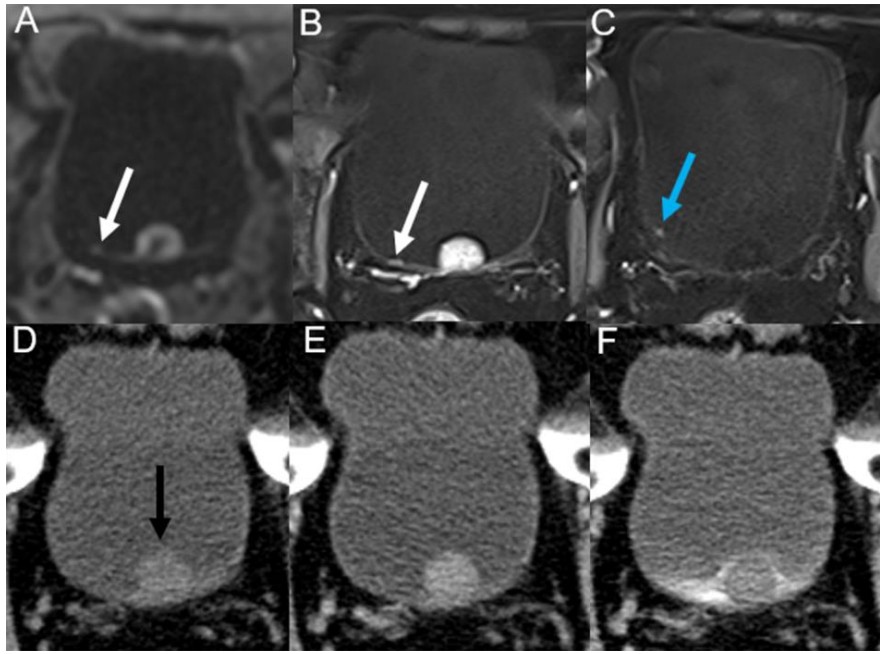

**Figure 3.** A 71-year-old man underwent trans-urethral resection of BC 42 months ago. MRI (**A**–**C**) and MDCT (**D**–**F**) show a papillary mass (black arrow) in median posterior wall of the bladder. A micro-nodule (white arrow) in the right posterior wall is visible on DWI (**A**) and DCE (**B**) images. Another tiny polypoid lesion (blue arrow) in the right posterolateral wall is visible on another layer of DCE-MRI (**C**). None of the two small lesions is visible on unenhanced (**D**), nephrographic (**E**), excretory (**F**) phases of MDCT. Postoperative pathology of all the lesions were non-muscle-invasive low-grade papillary urothelial carcinoma.

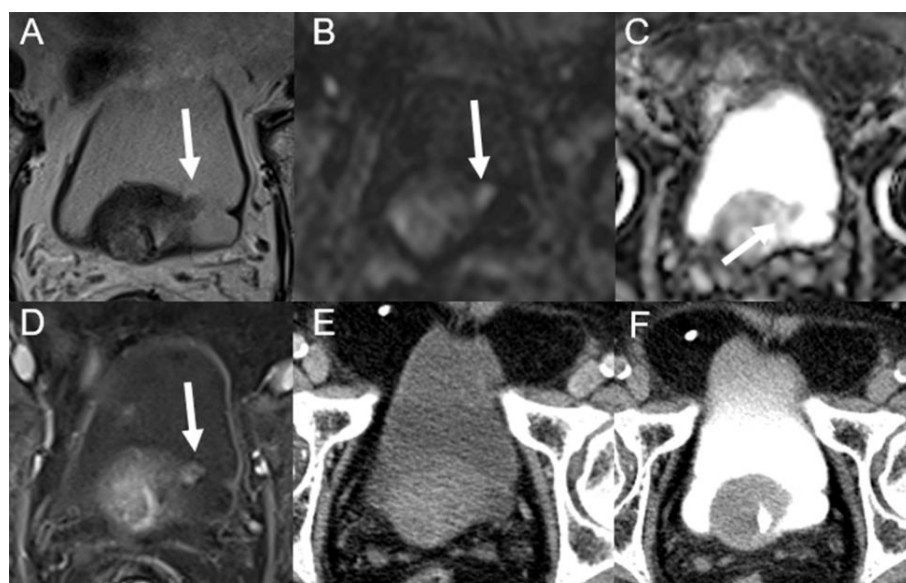

**Figure 4.** An 86-year-old man underwent trans-urethral resection of BC 48 months ago. T2WI (**A**), DWI (**B**), apparent diffusion coefficient (**C**) and DCE (**D**) images show a papillary mass (arrow) with a diameter of 0.9 cm in the left posterior wall of the bladder. However, multidetector computed tomography (only nephrographic (**E**) and bladder filling (**F**) phases are shown in the figure) shows no definitely noticeable lesion. Postoperative pathology confirmed non-muscle-invasive low-grade papillary urothelial carcinoma.

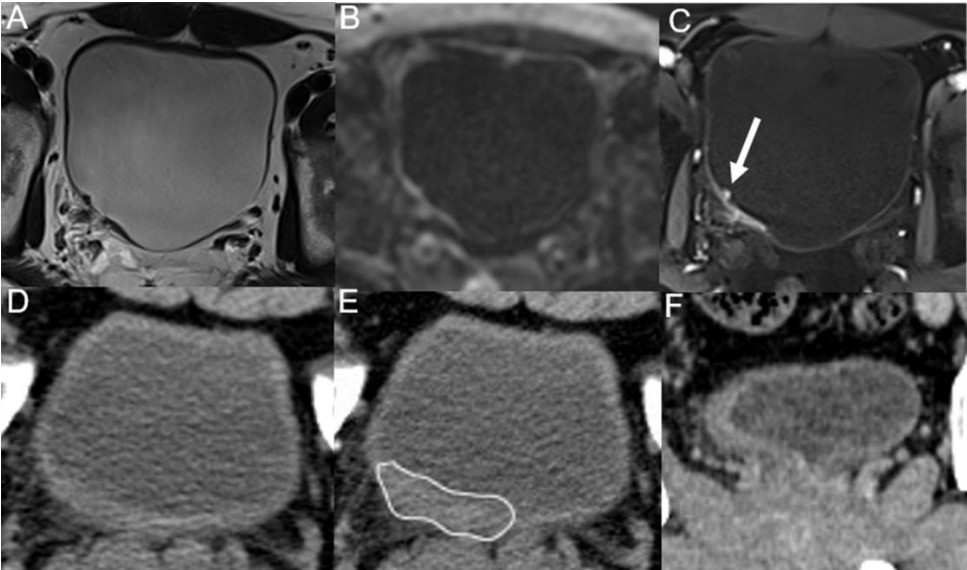

**Figure 5.** A 43-year-old man underwent trans-urethral resection of BC 5 months ago. T2WI (**A**), DWI (**B**) and DCE (**C**) images show irregular thickening of the right posterior wall of the bladder, with nodule-like enhancement (arrow) visible on DCE image, and all observers diagnosed it as tumor recurrence. Unenhanced (**D**), corticomedullary (**E**) phases and coronal reconstruction (**F**) images of multidetector computed tomography only show irregular wall thickening (line), and all observers diagnosed it as inflammation. Postoperative pathology confirmed it as low-grade malignant potential urothelial papillary tumor.

**Table 4.** MRI and MDCT diagnostic accuracy of different tumor morphologies in 56 patients with recurrent BC.

| Morphological Classification | MRI | MDCT |
|---|---|---|
| Nodular masses | 100% (38/38) | 92.1% (35/38) |
| Irregular wall thickening | 90.0% (9/10) | 30.0% (3/10) |
| Smooth wall thickening | 25.0% (2/8) | 0% (0/8) |

Tumors were stratified according to tumor morphology in cystoscopy and resected gross specimens.

## 4. Discussion

The study compared the diagnostic value of MRI and MDCT in detecting tumor recurrence after TURBT. The results showed that MRI has a good potential to differentiate tumor recurrence from benign treatment-related changes. MRI could better evaluate the curative effect of TURBT and was able to detect small lesions that were invisible on MDCT.

Although imaging examinations cannot replace cystoscopy, we must recognize that it is impractical to perform cystoscopy in all patients after TURBT. In clinical practice, MRI or CT examination may avoid invasive risks and prolong the interval between cystoscopy, which may encourage patients to be more compliant with follow-up. Therefore, it is important to evaluate the imaging diagnostic performance in differentiating tumor recurrence from postoperative benign changes.

Our research showed that imaging diagnosis is highly reproducible among experienced observers (κ > 0.90). Although CT is convenient and fast, it may cause X-ray radiation damage to patients. The recurrence detection rate of CT is not high. A multicenter retrospective study reported that the sensitivity, specificity and accuracy of enhanced CT for detecting tumor recurrence were 86%, 59% and 81%, respectively [14]. Our study had higher specificity (80.6%) but lower sensitivity (67.9%) and accuracy (73.7%). This may be because their research subjects were patients with suspicious recurrent urothelial carcinomas, while our research subjects included patients who were routinely followed up. Moreover, CT has some limitations in detecting small lesions and urethral diseases owing to relatively poor tissue contrast. Therefore, early assessment of tumor recurrence may be delayed. Two observers missed 10 cases of small lesions (<1 cm) on MDCT. Wang et al. reported that nearly all tumors ≥1 cm were detectable, while a third of tumors <1 cm were not detectable by MDCT urography [7]. MDCT missed one case of urethral metastasis in our study. This finding is consistent with a previous study that reported that MRI can depict urethral metastasis more precisely than CT after radical cystectomy [15].

Some studies indicated that MRI is highly reliable in distinguishing post-TURBT inflammatory changes from tumor recurrence [11,16]. The sensitivity, specificity and accuracy of DWI-MRI in detecting recurrence were 100%, 81.8% and 92.6% by Wang [11], and 91.6%, 91.3% and 91.5% by El-Assmy [16]. Their studies used DWI alone or combined with DCE-MRI with a slice thickness of 4 or 5 mm, respectively. Our study used mpMRI with a thinner slice thickness (3.5 mm) for a comprehensive diagnosis, and the sensitivity, specificity and accuracy were 87.5%, 94.4% and 90.2%, respectively. The lower sensitivity may be due to the difficulty of identifying recurrent lesions with smooth wall thickening.

Inflammatory edema, fibrosis and scar tissue secondary to TURBT and intravesical instillation can morphologically manifest as mural nodules or irregular wall thickening, thus simulating tumor recurrence [10,11,16,17]. It is difficult to distinguish from flat or diffuse tumors that infiltrate the bladder wall, which is the difficulty of imaging diagnosis.

To evaluate the influence of tumor morphology on imaging diagnosis, we classified recurrent patients into three groups. Most cases were nodular masses with a high accuracy. However, the accuracy decreased as lesion size decreased. Postoperative inflammation and fibrosis can be enhanced due to peri-tumoral neovascularization, which may persist for many years as an inflammatory change [11,17]. Among patients with irregular wall thickening, six cases were misdiagnosed as inflammation by MDCT due to focal wall thickening with homogeneous enhancement. These lesions showed high SI on DWI, intermediate or low ADC values, early obvious enhancement and slightly decreased

enhancement in the delayed phase on DCE-MRI. Because MRI has the advantages of high tissue contrast resolution and multiparametric imaging, DWI especially can reflect molecular diffusion restriction in malignant tissue. Previous studies [18] reported that an inflammatory and thickened bladder wall may be mistaken for recurrence if manifested as high SI on DWI, but this change showed no diffusion restriction. Therefore, DWI combined with ADC was the main sequence to identify recurrence and benign treatment-related changes [18]. We found that DCE-MRI also has some differential effects. One patient with nodule-like enhancement increased our confidence in the diagnosis of tumor recurrence.

Both MRI and MDCT misdiagnosed two cases of inflammation with a thickened bladder wall as recurrences. We found that the lesions were located at the site of previous resections. Moreover, MDCT misdiagnosed one papilloma as BC, whereas MRI showed no diffusion restriction. In addition to postoperative inflammatory changes that can mimic cancers, false positives may be related to a heightened concern for tumors given the prior history of urothelial cancer [19].

It is difficult to identify smooth wall thickening that shows only mucosal changes on cystoscopy without any visible papillary component, making it extremely difficult to detect with imaging [13]. The two cases with suspected positive MRI findings showed only linear enhancement relative to the remaining bladder wall, with high or slightly high SI on DWI and slightly low or intermediate ADC values. Notably, two papillomata (0.3–0.5 cm) were found intraoperatively in one patient with negative imaging findings. Therefore, the imaging diagnostic value of such patients is limited and requires cystoscopy.

This study has some limitations. First, inherent bias may exist because this was a single-center retrospective study with a limited patient cohort. Therefore, further external validation with larger databases from multiple centers is required to validate our results. Second, the large proportion of recurrent patients with nodular masses may make our data ideal to some extent, and more cases of the other two types need to be collected to validate our conclusions.

## 5. Conclusions

Compared with MDCT, MRI had a higher accuracy in detecting BC recurrence early, especially for nodular masses and irregular wall thickening, and could better differentiate tumor recurrence from benign treatment-related changes.

**Author Contributions:** Conceptualization, M.L., Y.W. (Yiqian Wang) and W.Z.; methodology, Y.W. (Yiqian Wang) and W.Z.; software, Y.W. (Yiqian Wang) and W.Z.; validation, M.L., Y.W. (Yiqian Wang) and W.Z.; formal analysis, Y.W. (Yiqian Wang), W.Z. and S.C.; investigation, Y.W. (Yiqian Wang) and W.Z.; resources, M.L., S.C. and Y.W. (Yongbao Wei); data curation, W.X. and S.C.; writing—original draft preparation, Y.W. (Yiqian Wang) and W.Z.; writing—review and editing, M.L.; visualization, W.X. and Y.W. (Yongbao Wei); supervision, M.L. and W.X.; project administration, M.L. and W.X.; funding acquisition, Y.W. (Yongbao Wei) and M.L. All authors have read and agreed to the published version of the manuscript.

**Funding:** This research was funded by the Joint Funds for the Innovation of Science and Technology, Fujian Province, grant number 2017Y9064; the Middle-aged Backbone Project Health and Family Planning Commission, grant number 2017-ZQN-13; and the Fujian Natural Science Foundation, grant number 2021J05177.

**Institutional Review Board Statement:** The study was conducted in accordance with the Declaration of Helsinki, and approved by the Institutional Review Board of Fujian Provincial Hospital (protocol code K-03-115 and date of approval 1 December 2013).

**Informed Consent Statement:** Informed consent was obtained from all subjects involved in the study.

**Data Availability Statement:** The data presented in this study are available on request from the corresponding author. The data are not publicly available due to privacy.

**Acknowledgments:** We appreciated greatly all the participants included in the study.

**Conflicts of Interest:** The authors declare no conflict of interest.

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
