# Peer review of "Feasibility of Early Evaluation for the Recurrence of Bladder Cancer after Trans-Urethral Resection: A Comparison between Magnetic Resonance Imaging and Multidetector Computed Tomography"

_tomography, doi:10.3390/tomography9010003_

Round 1

Reviewer 1 Report

Feasibility of Early Evaluation for the Recurrence of Bladder Cancer after Trans-Urethral Resection: A Comparison between Magnetic Resonance Imaging and Multidetector Computed Tomography

The paper is well written and requires very few corrections.

- Line 150 : Please correct the number of observers

- Line 235 : Please rephrase

- Figure 4 : A late phase picture of the MDCT with contrast-filled bladder might be useful

Regarding the morphological classification of the findings it might be useful to specify more in details the difference between irregular and smooth wall thickening.

Author Response

Point 1: - Line 150 : Please correct the number of observers.

Response 1: We are very sorry for our incorrect writing. The correct number of observers is two.

Point 2: - Line 235 : Please rephrase

Response 2: We modify the expression of the sentence “6 cases of irregular wall thickening” to “6 cases of focal wall thickening”. Can the modification meet your requirements?

Point 3: - Figure 4 : A late phase picture of the MDCT with contrast-filled bladder might be useful

Response 3: We use bladder filling phase image to replace unenhanced image.

Point 4: Regarding the morphological classification of the findings it might be useful to specify more in details the difference between irregular and smooth wall thickening.

Response 4: Originally, we wrote that the morphological classification of recurrent patients was based on pathology. This expression is not rigorous. In fact, it was based on what the clinicians observed in cystoscopy and resected gross specimens during surgery, which has been revised in manuscript. We have supplemented the morphological classification standard in “2.5 reference standard”. The details are as follows: nodular masses referred to lesions with papillary or cauliflower shape; irregular wall thickening referred to lesions that were focal wall thickening compared with the adjacent bladder wall, or diffuse wall thickening in cystoscopy, and the thicknesses of resected gross specimens were greater than or equal to 3 mm; smooth wall thickening referred to lesions with smooth or slightly gross mucosa in cystoscopy, and the thicknesses of resected gross specimens were less than 3 mm.

Reviewer 2 Report

Wang et al. present an interesting topic of the diagnosing power between MRI and MDCT in recurrent patients of bladder cancer, after TUR or combined with chemotherapy. They did a comprehensive analysis one both MRI and TUR images of the 96 patients after TUR therapy. It turns out MRI might have better performance in determining the recurrent tumors at a much early stage. In general, this study is properly designed, and the results are clear and sound, and the references are relevant and updated. However, ROCs in Figure 2 are not curves. Is there only one data point for each curve?

Author Response

Point: Wang et al. present an interesting topic of the diagnosing power between MRI and MDCT in recurrent patients of bladder cancer, after TUR or combined with chemotherapy. They did a comprehensive analysis one both MRI and TUR images of the 96 patients after TUR therapy. It turns out MRI might have better performance in determining the recurrent tumors at a much early stage. In general, this study is properly designed, and the results are clear and sound, and the references are relevant and updated. However, ROCs in Figure 2 are not curves. Is there only one data point for each curve?

Response: We set tumor recurrence as positive and non-recurrence as negative, and assumed positive = 1, negative = 0 in SPSS software. Because there are only two values (1 and 0), the ROC curves obtained are as shown in Figure 2. An example may be more helpful for your understanding. It is assumed that only 5 patients are enrolled in this study. The imaging diagnosis and pathological results are shown in the following table. The ROC curves obtained by the statistical software will also be as shown in Figure 2, and each curve only has one data point.

Patients MRI MDCT Pathology
1 Positive→1 Negative→0 Positive→1
2 Positive→1 Positive→1 Positive→1
3 Positive→1 Positive→1 Positive→1
4 Negative→0 Positive→1 Positive→1
5 Negative→0 Positive→1 Negative→0